# Effective Initial Treatment of Diffuse Pulmonary Lymphangiomatosis with Sirolimus and Propranolol: A Case Report

**DOI:** 10.3390/medicina57121308

**Published:** 2021-11-29

**Authors:** Ieva Dimiene, Kristina Bieksiene, Jurgita Zaveckiene, Mindaugas Andrulis, Daiva-Elzbieta Optazaite, Neringa Vaguliene, Marius Zemaitis, Skaidrius Miliauskas

**Affiliations:** 1Department of Pulmonology, Medical Academy, Lithuanian University of Health Sciences, 44307 Kaunas, Lithuania; kristina.bieksiene@lsmu.lt (K.B.); neringa.vaguliene@lsmu.lt (N.V.); marius.zemaitis@lsmu.lt (M.Z.); skaidrius.miliauskas@lsmuni.lt (S.M.); 2Department of Radiology, Medical Academy, Lithuanian University of Health Sciences, 44307 Kaunas, Lithuania; jurgita.zaveckiene@lsmu.lt; 3Institute of Pathology, Ludwigshafen General Hospital, 67063 Ludwigshafen am Rhein, Germany; andrulism@klilu.de; 4Department of Radiology, Thoraxklinik at Heidelberg University Hospital, 69126 Heidelberg, Germany; elzbieta.optazaite@med.uni-heidelberg.de

**Keywords:** diffuse pulmonary lymphangiomatosis, sirolimus, propranolol, acute fibrinous organizing pneumonia, diagnosis, treatment

## Abstract

Diffuse pulmonary lymphangiomatosis (DPL), an exceptionally rare disease, mainly occurs in children and young adults of both sexes. Even though DPL is considered to be a benign disease, its prognosis is relatively poor. Because of its rarity, little guidance on diagnosis and treatment is available, which makes working with patients with DPL challenging for clinicians. We present here a case of a young man with DPL in whom treatment with sirolimus and propranolol rapidly achieved positive radiological and clinical effects.

## 1. Introduction

Lymphangiomatosis, an uncommon disease, is characterized by diffuse development of lymphangiomas in parts of the body that have lymphatics. When it occurs in the thorax, the condition is called diffuse pulmonary lymphangiomatosis (DPL) [1]. Histologically, lymphangiomas are a rare type of benign neoplasm. They are characterized by uncontrolled lymphatic vessel proliferation that is influenced by vascular endothelial growth factor (VEGF) [2,3]. Diagnosis of DPL is based on a combination of clinical, radiological (chest radiographs, computed tomography [CT] or magnetic resonance imaging) findings, results of pulmonary function tests (PFTs), and findings on bronchoscopy and immunohistochemical tests on appropriate tissue biopsies [4,5]. Diagnosis and treatment of DPL remain major challenges because the disease is exceptionally rare. Treatment choices are mostly based on case reports because no definitive guidelines for the management of DPL have yet been developed. According to such reports, sirolimus, bevacizumab, and propranolol [3,6,7,8,9,10,11] are the most commonly chosen and effective treatments for DPL. However, progressive DPL may require lung transplantation [12]. We here present a case of a 26-year-old man with DPL whose initial treatment with a combination of sirolimus and propranolol was effective.

## 2. Case Report

A 26-year-old man had been hospitalized repeatedly over approximately two years for dyspnoea, night sweats, and hemoptysis. He had a 7-year history of clinically insignificant pericardial effusion of unknown cause and a 2-year history of autoimmune thyroiditis with euthyroid status. Tonsillectomy had been performed at the age of 18. He had had bacterial meningitis at the age of three. His alkaline phosphatase and gamma-glutamyl transferase (GGT) concentrations had been significantly increased for two years; however, no cause of hepatitis had been identified. There was no family history of lung diseases. He had a healthy twin sister. The patient had a smoking history of 4 pack-years.

CT scans for the previous two years had shown progressive pleural effusions, parapleural lymphostasis, mediastinal lymphadenopathy, mediastinal soft tissue oedema, ground glass opacities (GGOs), and thickened pericardium (Figure 1). Repeated thoracenteses and pleural drainage had yielded chylous fluid.

Cardiac imaging showed a clinically insignificant pericardial effusion. The patient was thoroughly investigated for suspected lymphatic leakage, connective tissue diseases, lymphoproliferative disorders, immunodeficiency, chronic infections, and genetic disorders. However, no definitive diagnosis was reached.

Biopsies obtained by minimally invasive techniques were inconclusive, and therefore an open lung biopsy was performed, enabling a multidisciplinary team (MDT) to make a diagnosis of acute fibrinous organizing pneumonia (AFOP). A diagnosis of DPL was also considered by radiologists; however, there was insufficient evidence to support a definite diagnosis of this condition. Treatment with steroids (starting with intravenous pulse steroid therapy, followed by 64 mg oral methylprednisolone daily and gradually reducing to 4 mg daily) resulted in some temporary improvement in dyspnoea, hemoptysis, and radiological features of AFOP (Figure 2).

Seven months after initiation of treatment with methylprednisolone, the patient‘s dyspnoea and hemoptysis, however, progressed and pulmonary embolism (PE) was diagnosed. Anticoagulative therapy was initiated, starting with low-molecular-weight-heparin, which was followed by rivaroxaban. Thereafter, the patient was followed intensively by pulmonologists. His symptoms stabilized; however, repetitive thoracenteses were required, all of which yielded chylous fluid.

Two months after the diagnosis of PE, the patient was hospitalized in the Department of Pulmonology (Hospital of Lithuanian University of Health Sciences Kauno Klinikos) because of worsening dyspnoea and cough. A chest CT scan showed progression of bilateral hydrothoraxes, lymphostasis, compression of the lungs, multiple liquid–air interfaces in the right interlobar space, necrosis in the left basal segments (features of lung infarction), and multiple enlarged lymph nodes in the mediastinum.

At this point, the patient‘s medical history, archival lung tissue biopsies and radiological images were sent to a multidisciplinary team of experts in Germany (Thoraxklinik Heidelberg, Institute of Pathology, Klinikum Ludwigshafen) for a second opinion. There, experts in radiology and respiratory medicine concluded that the diagnosis was DPL. The archival lung biopsies were further processed, producing additional serial sections that were subjected to D2-40 immunohistochemistry to facilitate detection of lymphatic vessels. This resulted in identification of multiple foci of widely dilated lymphatic vessels that were predominantly localized in subpleural and peribronchial areas and greatly obscured by secondary changes, including thickened pleura and evidence of organizing broncho-pneumonia (Figure 3). These findings are characteristic of DPL, but had been difficult to detect because of the focal pattern of presentation and secondary inflammatory changes.

After discussing the findings and conclusions of the German specialists, our MDT accepted the suggested diagnosis of DPL and initiated treatment with sirolimus, a mechanistic target of rapamycin (mTOR) inhibitor. The initial dose of sirolimus was 2 mg daily. This was subsequently adjusted to maintain a target concentration of 10–15 mcg/L. Treatment with methylprednisolone for AFOP was continued because of persistent GGO on chest CT scans.

Three weeks after initiation of DPL treatment with sirolimus, the patient was hospitalized because of COVID-19 infection. This manifested as fever and progressive dyspnoea with no evidence of respiratory failure. A chest CT scan showed evidence of COVID-19 pneumonia and acute PE against a background of chronic lung disease and chronic PE with persistent bilateral pleural effusions. At the time of hospitalization, the patient’s SpO_2_ was normal (97%), C reactive protein (CRP) was slightly increased (16 mg/L), and there was no leukocytosis or neutrophilia. However, he did have lymphopenia (0.4 × 10^9^/L). Treatment with intravenous dexamethasone 8 mg once a day, bemiparin 10,000 IU once a day, propranolol 40 mg twice daily and sirolimus 1 mg once a day was initiated during this hospitalization. The patient did not develop respiratory failure, his CRP returned to within the normal range (≤5.0 mg/L) and the only persistent abnormality in his full blood count was lymphopenia. He was discharged from hospital after 7 days of treatment.

Since then, the patient has been taking apixaban 5 mg twice daily, methylprednisolone 4 mg once a day, propranolol 40 mg twice daily, sirolimus 1 mg once a day, and torsemide 20 mg as needed for recurrent peripheral oedema. Metoprolol, which had been prescribed earlier for tachycardia, was ceased and propranolol substituted because of the latter’s inhibitory effect on proliferation of lymphangiomas [10]. The patient also underwent pulmonary rehabilitation for 3 weeks after recovery from COVID-19 and acute PE.

After 3 months of treatment with sirolimus and propranolol, there was clinical and radiological evidence of improvement, with reduction of dyspnoea from 5 to 3 on the BORG CR10 scale [13] and significant regression of interstitial lesions, pleural effusions, and GGOs on chest radiographs (Figure 4). However, the patient’s PFTs did not alter significantly.

## 3. Discussion

The extreme rarity of DPL hindered diagnosis of this condition in our patient. Although his symptoms (dyspnoea and hemoptysis) and radiological findings (pleural effusion, parapleural lymphostasis, mediastinal lymphadenopathy and mediastinal soft tissue oedema, and ground glass opacities) were consistent with the eventual diagnosis of DPL, the correct diagnosis was made only by pathological examination of lung tissue obtained by open biopsy and repeated review of that biopsy tissue by multiple specialists. Unfortunately, it took approximately 2 years to arrive at the diagnosis of DPL.

According to Radhakrishnan et al., diagnosis of DPL is based on a combination of radiological findings, clinical features, and results of pulmonary function tests, lung biopsy being an ancillary means of making this diagnosis [14]. Because there are very few published studies on DPL, data on its clinical course and prognosis are available only from previous case reports. Multiple medications for DPL have been tried, including bevacizumab, propranolol, sirolimus, interferon-alfa 2b, glucocorticoids, bisphosphates, thalidomide [5,10], and sildenafil [15]. Other forms of treatment, such as surgery, sclerotherapy, radiotherapy, and lung transplantation have also been reported [12,16,17,18]. Additionally, dietary treatments such as medium-chain triglycerides and high-protein diets can reportedly have positive effects in patients with DPL [19]. The most recent case reports have shown that sirolimus, an mTOR inhibitor [8,11,20] or bevacizumab, a vascular endothelial growth factor (VEFG) inhibitor [18,21] may be quite effective, improving lung function and achieving good clinical and radiological responses without causing serious adverse effects. Treatment with propranolol has also been shown to be effective in the management of DPL [7].

In our case, bevacizumab was contraindicated because of repeated episodes of hemoptysis. Our first choice was therefore sirolimus; we aimed to maintain the blood concentration between 10 and 15 µg/L. We added propranolol for the following two reasons: to control the tachycardia that had previously been treated with metoprolol and to support the treatment of DPL via reduction of levels of VEGF [10]. Because of his repeated chylothoraxes, we also recommended a medium-chain triglyceride and high protein diet. This combination of optimal pharmacological and non-pharmacological treatment measures led to a positive initial clinical and radiological response.

One limitation of the present report is that we only evaluated the short-term response to the selected treatment, whereas other authors published the results of 12 and 21 months of treatment [8], and 36 months of treatment [19]. Moreover, the only post-initiation of treatment chest CT scan that we performed was carried out during the patient’s COVID-19 infection. Thus, our patient’s radiological response to DPL treatment response is based only on plain chest radiographs. Our patient is continuing the treatment and we plan to publish further results in the future.

## 4. Conclusions

The rarity of DPL means that no guidelines are available on its management, which makes arriving at the correct diagnosis and prescribing the optimal treatment challenging. However, recent case reports and case series have presented several alternatives for treating DPL. Our patient had positive short-term clinical and radiological responses to treatment with sirolimus and propranolol.

## Figures and Tables

**Figure 1 medicina-57-01308-f001:**
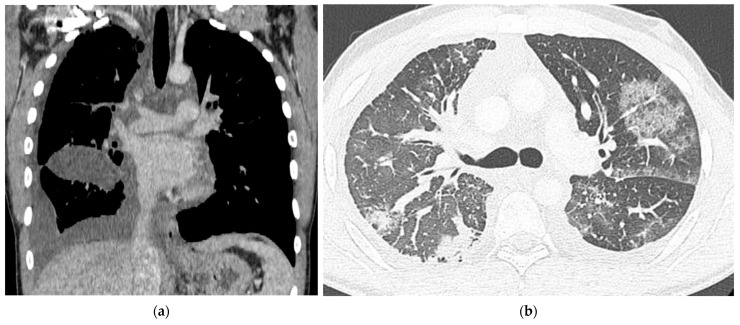
Chest CT images prior to establishing a diagnosis and initiation of treatment showing pleural effusions, parapleural lymphostasis, mediastinal lymphadenopathy and oedema (**a**) and GGOs (**b**).

**Figure 2 medicina-57-01308-f002:**
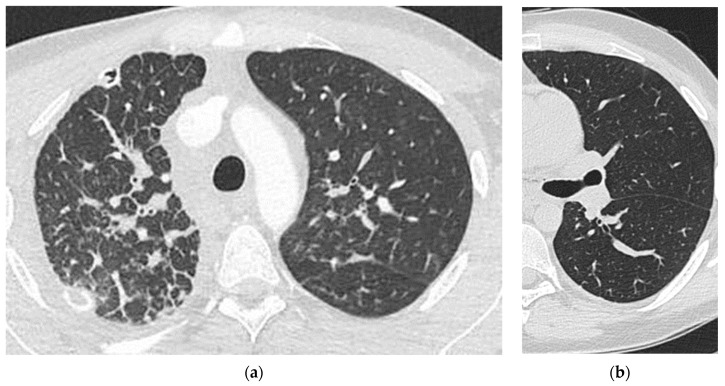
Chest CT images after initiation of steroid treatment showing residual lymphostasis, mainly in the upper lobes (**a**) and significantly less marked features of AFOP (**b**).

**Figure 3 medicina-57-01308-f003:**
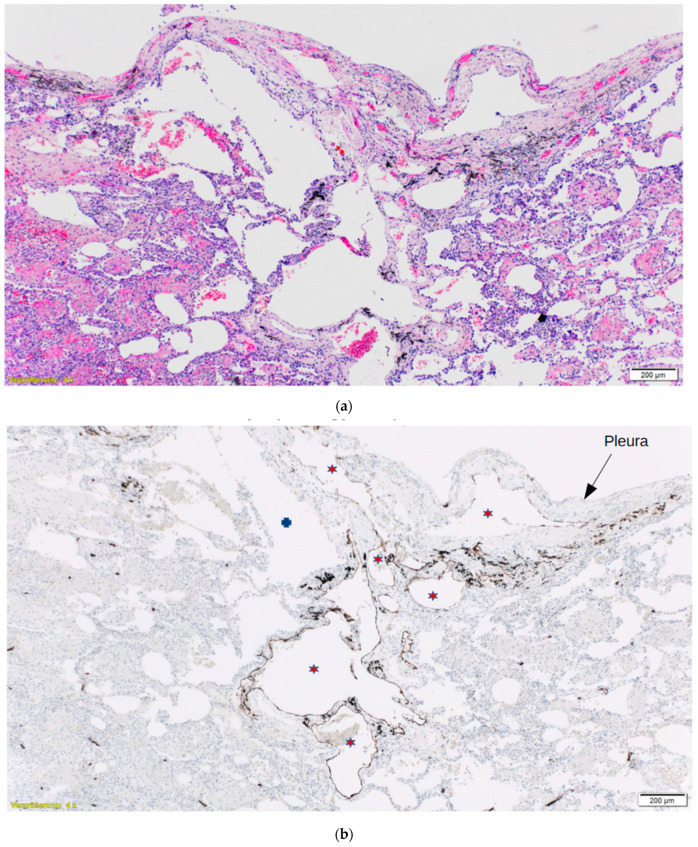
Hematoxylin–eosin (H&E)-stained (**a**) and D2-40 immunohistochemistry stained (**b**) photomicrographs of lung tissue obtained by open biopsy demonstrating multiple foci of dilated lymphatic vessels (★) in subpleural locations. Pleura is marked with the arrow (↓). The H&E section also depicts secondary changes, including pleural thickening, organizing pneumonia, and bullous alveoli (✠), all of which masked the main pathological process.

**Figure 4 medicina-57-01308-f004:**
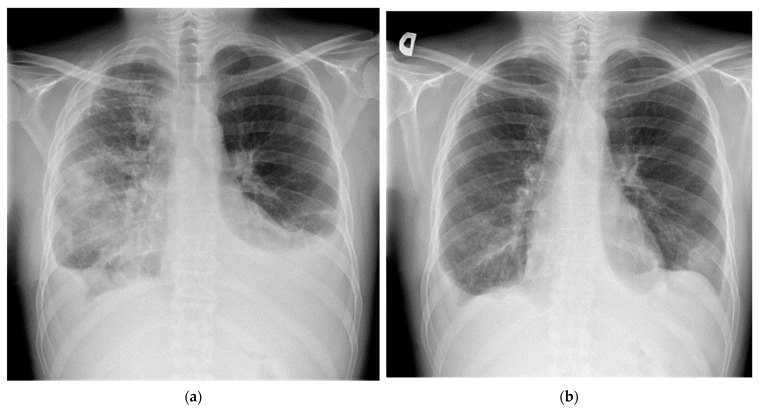
Chest radiographs taken before initiation of DPL treatment (**a**) and after 3 months of that treatment (**b**). Bilateral reduction in lymphostasis, pleural effusion, and GGO is apparent. There is an area of infarction in the left lower lobe.

## Data Availability

Not applicable.

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
