# Peer review of "Effective Initial Treatment of Diffuse Pulmonary Lymphangiomatosis with Sirolimus and Propranolol: A Case Report"

_medicina, 2021, doi:10.3390/medicina57121308_

Round 1

Reviewer 1 Report

Diffuse pulmonary lymphangiomatosis (DPL) is an exceptionally rare disease. .This manuscript presents a 26-year-old man with DPL whose initial treatment with a combination of sirolimus and propranolol was effective. In fact, the management of DPL have yet been developed. The patient’s treatment with methylprednisolone was evaluated for seven months,but treatment with a combination of sirolimus and propranolol was only evaluated for 3 months.

Author Response

Dear Reviewer,

Thank You very much for Your comments.

We would like to explain that methylprednisolone was prescribed for the treatment of acute fibrinous organising pneumonia (AFOP) before the diagnosis of diffuse pulmonary lymphangiomatosis (DPL) was obtained. Treatment with methylprednisolone led to regression of AFOP's features, however, chylothorax and lymphostasis progressed, the biopsy samples were reviewed and diagnosis of DPL was confirmed. Chylothorax reduced only after initiation of sirolimus and propranolol.

In this manuscript we would like to arise the effectivenes of combination of sirolimus and propranolol in the treatment of DPL.

Yours sincerely,

Ieva Dimiene

Reviewer 2 Report

Since diffuse pulmonary lymphangiomatosis is a rare disease with diverse medical findings, it cannot be helped that it took a long time to diagnose. I think this manuscript will contribute to the future treatment of diffuse pulmonary lymphangiomatosis cases.

Author Response

Dear Reviewer,

We are very grateful for Your time and positive comments.

Yours sincerely,

Ieva Dimiene

Reviewer 3 Report

This is a very interesting and rare case of Diffuse Pulmonary Lymphangiomatosis (DPL), which has been treated successfully with Sirolimus and Propranolol.

@@Some questions need to be clarified as following

# Introduction:

  1. Two important case reports regarding sirolimus in treating DPL has not been listed and discussed in the introduction part.
  2.  reference: Respir Med Case Rep. 2020 Feb 1;29:101014 & Journal of the Belgian Society of Radiology. 2018;102(1):64.

#. Case reports:

  1. Pulmonary function test is the gold standard to make the diagnosis of DPL and is also an important clinical feature to monitor disease progression, however, the manuscript did not show the spirometry results and serial follow-up in this patient. In addition, this case can not follow-up by chest CT scan, therefore, we need the serial lung function test data to support the treatment effect of Sirolimus and Propranolol.
  2. The typical presentation of DPL on immunohistochemical stain is positive of D2-40, CD31 due to the proliferations of endothelial cells and negative for HHV-8. Could you provide the findings of CD31 and HHV-8 in the case?

#. Discussion:

  1. Previous studies suggest the treatment course of Sirolimus required several months to years for patients with DPL. In a case report published in 2018 by C. Ernotte et al, the patient has been treated with Sirolimus for 4 years. Another case presented by Gurskyte et al in 2020 showed the slightly decreased in size of mediastinal lesions after 21 months of treatment.
  2. In this case, the author mentioned about a great improvement on clinical and CXR of patient after 3 months course of treatment with sirolimus and propranolol. However, no significant improve on lung function test. Did you performe right side thoracocentesis for this patient? If no pleural drainage has been done, is there any possible mechanism regarding sirolimus on pleural effusion?

Author Response

Dear Reviewer,

We are very grateful for Your accurate comments and questions.

The answers to Your questions are written below:

-Introduction:

Thank You for useful remarks about the case reports that should be used in the paragraph. We will add them to the corrected version.

-Case reports:

1. We did not perform serial pulmonary function tests (PFT's) due to repeating haemoptysis and pulmonary embolism.

As we have mentioned in the text, the results of PFT's that we performed did not show a significant difference before and in the course of the treatment. However, clinical and radiological effects were observed.

In our case clinical and radiological features as well as biopsy results were the main components for the diagnosis of DPL.

2. Unfortunately, we cannot provide You with CD31 and HHV-8 findings. Our pathologists used D2-40 immunohistochemistry.

-Discussion:

The clear action of mechanism is not clear, however, relying on the scientific literature, sirolimus and propranolol inhibits the growth of lymphatic vessels which can lead to reduction of chylothorax. In this case we did not perform the drainage of right pleura. We consider that the reduction of chylothorax is due to sirolimus and propranolol.

I hope my answers helped clarifying the questions.

Yours sincerely,

Ieva Dimiene